# Evaluation of Bacterial Diversity and Quality Features of Traditional Sichuan Bacon from Different Geographical Region

**Song Wang [1,2], Xingjie Wang [1], Wanshu Pan [2], Aiping Liu [1], Shuliang Liu [1,*], Yong Yang [1] and Likou Zou [3]**

1 College of Food Science, Sichuan Agricultural University, Ya'an 625014, China; wangsongybxy@163.com (S.W.); wxjseeseven@163.com (X.W.); aipliu@outlook.com (A.L.); yangyong676@163.com (Y.Y.)
2 Faculty of Agriculture, Forestry and Food Engineering, Yibin University, Yibin 644000, China; panwanshu93@163.com
3 College of Resources, Sichuan Agricultural University, Ya'an 625014, China; zoulk124@163.com
* Correspondence: lsliang999@163.com; Tel.: +86-835-2882187





**Featured Application: This work can help us understand the role of bacterial communities in Sichuan bacon more deeply and accurately.**

**Abstract:** Sichuan bacon is one of the most popular types of Chinese bacon in the domestic market. High-throughput sequencing was used to characterize the bacterial diversity of 39 Sichuan bacon samples collected from 3 geographical regions. The results showed that the bacterial diversity of Sichuan bacon in different regions demonstrated certain specificity as well as similarity, and the shared OTUs were close to 81% of the total number in the basin group, mountain group, and plateau group. At the genus level, *Staphylococcus* is the most dominant genus among the three groups, covering 26.7%, 20.6%, and 22.7%, respectively. Beta diversity shows significant differences in bacterial compositions in different geographic regions, especially for *Pseudomonas*, *Brochothrix*, *Lactobacillus*, *Lactococcus*, and *Enterococcus*. Meanwhile, some physicochemical characteristics were analyzed, and a significant difference ($p < 0.05$) among the three regions was shown in the Aw, pH, and nitrite content, which were significantly correlated with undesired bacteria. This study provides insights into the understanding of the role of bacterial communities in the microbial safety and quality improvement of Sichuan bacon.

**Keywords:** Sichuan bacon; bacterial diversity; high-throughput sequencing; geographical regions; specificity; undesired bacteria

## 1. Introduction

Chinese bacon, known as "larou" in China, is one of the most popular traditional dry-cured meats in the domestic market [1]. Southern China is an extremely suitable area for bacon production due to its geographical and climatic conditions, especially in the Sichuan province, which includes the basin plain (basin), the Qinghai-Tibet Plateau (plateau), and the mountainous transition zone (mountains), with altitude differences of more than 3000 m [2]. Traditional Sichuan bacon is produced in December on the lunar calendar. After slaughtering, the pork belly or hind leg meat is sampled and cut into uniform strips. The strips are evenly wiped with salt, layered in a ceramic tank, salted for 7–10 days, and turned over 2 or 3 times during this period. The meat strips are cleaned with warm water and air-dried naturally following the curing. Finally, all the strips are continuously hung outdoors or intermittently smoked by a cooking stove for ripening for about 3 weeks until becoming the final products. The entire production cycle is approximately 30–35 d. In Sichuan province, there are more than 2000 small- and medium-sized meat processing enterprises, and cured meat products such as Sichuan bacon and sausage account for 80% of the total meat product consumption. In the meantime, there are more than 10 million

families in Sichuan province, and each family will make Sichuan bacon in December on the lunar calendar every year. Sichuan bacon together with Jinhua ham, Cantonese sausage, and Nanjing dry-cured duck are the most famous traditional meat products in China.

In modern fermented meat production, the use of starter cultures, mainly constituted of lactic acid bacteria (LAB) and coagulase-negative staphylococci (CNS), is becoming more frequent to guarantee safety and to standardize product properties. For example, meat products with the addition of starter cultures have better flavor, quality, and safety properties than products without them [3,4]. Unfortunately, Sichuan bacon is still produced with traditional technologies and without selected starters, which means its production is more of an art, depending on the skill and experience of the meat manufacturer rather than a process fully based on scientific and technological means [5]. Although the use of commercially available starters may result in a loss of the desirable sensory characteristics found in artisan fermented meat with less flavor [6], industrialized production of starter cultures is a consequence of the gradual shift in traditional meat production from small, local producers to large-scale processing plants [7].

Previous studies showed that the most promising strains for starter cultures are those which are isolated from the indigenous microbiota of traditional products [8–10]. These strains are well adapted to the meat environment and the specific manufacturing process and thus are capable of dominating the microbiota of products [11]. Fermented meat products must be manufactured with the addition of autochthonous microorganisms not only for the preservation of regional typicality but also for avoiding risks to consumers' health [12]. In other words, the autochthonous microbiota of fermented meat products must be analyzed in detail, because commercial starter cultures usually originate either from substrates or from the processes in which they are applied [5]. In order to protect the traditional aspects of these products and select the autochthonous starter cultures to be used, it is essential to understand the microbial dynamics during fermentation [7].

However, the research on traditional Sichuan bacon has rarely been reported in the literature [13], and the bacterial community dynamics are still unknown. Although some strains have been studied intensively in laboratory patterns, the bacterial communities have yet to be completely clarified in fermented meat because of the drawback of the culture-dependent method [10]. Now, high-throughput sequencing (HTS) methods are particularly suitable for research of fermentation food due to the more accurate and in-depth information attained compared with traditional methods involving the microbiota, especially in spontaneous fermentation matrixes such as cheese, sourdough, kimchi, as well as cured meat products [14–17].

In this study, we identified the bacterial diversity of Sichuan bacon sampled from three altitude zones in Sichuan province of China by an HTS approach and distinguished the general features of bacons through its physicochemical characteristics. Our results are expected to provide a comprehensive comparison of the bacterial diversity profiles among three geographic regions and insights into the significant differences of the complex microbial compositions in Sichuan bacon of different areas. In the meantime, the undesired bacteria community was checked by HTS technology to evaluate the microbial safety of Sichuan bacon. These detailed data can help us understand the role of bacterial communities in Sichuan bacon more deeply and accurately.

## 2. Materials and Methods

### 2.1. Sample Collection

Thirty-nine samples of traditional Sichuan bacon were randomly collected from 27 individual households in 8 cities across the Sichuan province of China, which were classified into three groups: basin (a total of 14 samples, numbered BA1–BA4, BB1–BB5, and BC1–BC5), plateau (a total of 10 samples, numbered PA1–PA5, and PB1–PB5), and mountains (a total of 15 samples, numbered MA1–MA5, MB1–MB5, and MC1–MC5), according to their geographical origins (Table 1). All samples, with production cycles of 35 ± 2 d, were collected between 1 March 2018 and 5 March 2018. Each sample was vacuum-packaged in

situ and shipped back to the laboratory with ice packs. Each sample was minced and mixed after removing the visible skin, fat, and connective tissue in the ultra-clean workbench. Immediately, 10 g of minced samples were packed into a sterile homogenized bag for DNA sequencing. The remaining part of each sample was packaged for other tests and stored at −80 °C.

**Table 1.** The information of the sampling locations.

| Region | City | Encode | Average Temperature (°C) [A] | Average Relative Humidity (%) [A] | Altitude (m) |
|---|---|---|---|---|---|
| Basin | Suining<br>Leshan<br>Chengdu | BA<br>BB<br>BC | 7~8 °C | 75~80% | <500 m |
| Mountains | Yibin<br>Ya'an<br>Xichang | MA<br>MB<br>MC | 9~10 °C | 75~80% | 500~2000 m |
| Plateau | Kangding<br>Ma'er kang | PA<br>PB | −1~0 °C | 50~55% | >2500 m |

All data come from the Sichuan Statistical Yearbook. [A] Results are expressed as a range of 5 years on average (2014–2018), and the data of each year only consist of December of the current year and January of the following year (the production period of Sichuan bacon).

### 2.2. Physiochemical Characteristics of Sichuan Bacon

The water activity (Aw) values were determined as previously described by Wang et al. [18]. One gram of minced sample was measured at 25 °C using a water activity meter (HD-3A, Huake Instrument and Meter Co., td, Wuxi, China). The pH values were determined as described by Jin et al. [19] and were recorded using a pH meter (PHSJ-3F, INESA Scientific Instrument Co., Ltd., Shanghai, China). The salt content was scored as chloride and was quantified according to ISO 1841-1:1996(E). The peroxide value (POV) and nitrite content were measured as described by Jin et al. [20]. All experiments were performed in triplicate, and the results represented the mean values of three independent experiments.

### 2.3. DNA Extraction and Amplicon Sequencing

The total genome DNA from the samples was extracted using the CTAB/SDS method. The DNA concentration and purity were monitored on 1% agarose gels. The V4 region of the bacterial 16S rRNA gene was amplified with the universal primers 515F (5′-GTGCCAGCMGCCGCGGTAA-3′) and 806R (5′-GGACTACHVGGGTWTCTAAT-3′). The PCR mixture (30 μL) consisted of 15 μL of the Phusion® High-Fidelity PCR Master Mix (New England Biolabs), 0.2-μM primers, and 10 ng of template DNA. The amplification cycles were as follows: 98 °C for 1 min; 30 cycles (98 °C for 10 s, 50 °C for 30 s, and 72 °C for 60 s); and then a final extension at 72 °C for 5 min. PCR products were mixed with the same volume of 1× loading buffer (contained SYB green) and operated electrophoresis on 2% agarose gel for detection. Samples with bright main strips between 400–450 bp were chosen for further experiments. PCR products were mixed in equidensity ratios and then purified with a Qiagen Gel Extraction Kit (Qiagen, Hilden, Germany). Sequencing libraries were generated using a TruSeq® DNA PCR-Free Sample Preparation Kit (Illumina, San Diego, CA, USA) following the manufacturer's recommendations, and index codes were added. The library quality was assessed on a Qubit@ 2.0 Fluorometer (Thermo Scientific, Waltham, MA, USA) and the Agilent Bioanalyzer 2100 system. At last, the library was sequenced on an IlluminaHiSeq2500 platform, and 250-bp paired-end reads were generated.

### 2.4. Sequence Analyses

Paired-end reads were assigned to samples based on their unique barcodes and truncated by cutting off the barcode and primer sequence. The paired-end reads were merged using FLASH (V1.2.7) to obtain raw tags. Then, quality filtering of the raw tags

was performed under specific filtering conditions to obtain the high-quality clean tags according to the QIME (V1.7.0) quality-controlled process. After chimeric sequences were identified and removed by the UCHIME algorithm (UCHIME Algorithm), effective tags were obtained and clustered into operational taxonomic units (OTUs) with a 97% similarity threshold. Taxonomy assignment of the OTUs was performed by comparing the sequences to the SILVA database.

The abundance information of the OTUs was normalized by the sample with the least sequences. A Venn diagram was illustrated to display the similarities and differences of the communities among the regions. Alpha diversity (Chaol and ACE richness estimators, Shannon and Simpson diversity indices, and Good's coverage) was used to analyze the bacterial diversity in a single sample. The beta diversity based on the Bray-Curtis and Unifrac distance was used to evaluate the differences between the microbial communities. Principal component analysis (PCA), principal coordinate analysis (PCoA), and non-metric multidimensional scaling (NMDS) were performed using QIIME software (Version 1.7.0). In the meantime, linear discriminant analysis (LDA) effect size (LEfSe) was applied to illustrate statistically significant biomarkers in each region with LDA > 4 [21].

### 2.5. Statistical Analysis

The collected data were subjected to one-way analysis of variance (ANOVA). Statistical significance in the group means was evaluated at $p < 0.05$ using Duncan's multiple range tests. Statistical analysis of the data was carried out in SPSS 22.0 (SPSS Inc., Chicago, IL, USA).

## 3. Results

### 3.1. Physicochemical Parameters of Sichuan Bacon

The results for the water activity (Aw), pH, salt content, nitrite content, and peroxide value (POV) of all samples from three different regions are shown in Table 2. The average values of the POV and salt content of three groups were consistent with previous reports [19], and no significant difference ($p \geq 0.05$) was observed among the groups. However, the POV and salt content of the basin group bacons were the highest, and those of the plateau group bacons were the lowest among the three groups. This phenomenon could be due to the fact that NaCl had a pro-oxidant effect on the lipids in cured meat products within the certain design range, especially for lipid primary oxidation [22].

The average Aw values of bacons from the mountains group and plateau group were 0.855 and 0.879, respectively, with no significant difference ($p \geq 0.05$). Conversely, the average Aw value of the basin group (mean = 0.814) was significantly ($p < 0.05$) lower than that of the other two groups. All samples were collected from different locations within only 7 days and were produced by similar processing times (about 35 d). However, dehydration of bacon in the basin group was more severe than for the other two groups. The reason for this might be that the free water of the samples evaporated more easily into the air in the basin region than in the other two regions during bacon processing. Another possible explanation might be that there were some slight differences in the production process of Sichuan bacon in different regions. As with the Aw, the average pH value of the basin group was significantly ($p < 0.05$) lower than that of the mountain and plateau groups. The metabolism of both the acid-producing bacteria in the initial curing period and the ammonia-producing bacteria in the later ripening period led to changing the pH of the meat matrix and further affecting the pH of the final product [23]. The average pH values of the basin, mountain, and plateau groups were 5.45, 5.68, and 5.92, respectively, which might indicate that there were some differences in the metabolic activity of those bacteria in the samples from the three regions.

Although no nitrite was added to the traditional Sichuan bacon production, a low content of nitrite in all samples was observed. Interestingly, the nitrite content of the mountains group was significantly ($p < 0.05$) higher than those of the basin and plateau groups. In the production of cured meat products, nitrate could be reduced to nitrite by

nitrate reductase from microorganisms such as *Staphylococcus* and then to nitric oxide (NO), which contributes to color development and microbial safety [24].

**Table 2.** Physicochemical indices in Sichuan bacon subjected to three groups of different regions.

| Parameters | Groups | | |
|---|---|---|---|
| | Basin (*n* = 14) | Mountains (*n* = 15) | Plateau (*n* = 10) |
| **Aw** | | | |
| Means ± SE | 0.814 ± 0.037 [b] | 0.855 ± 0.023 [a] | 0.879 ± 0.049 [a] |
| Range | 0.761~0.924 | 0.825~0.912 | 0.828~0.974 |
| **pH** | | | |
| Means ± SE | 5.45 ± 0.36 [b] | 5.68 ± 0.30 [ab] | 5.92 ± 0.25 [a] |
| Range | 5.02~6.03 | 5.27~6.48 | 5.59~6.44 |
| **Salt content [A]** | | | |
| Means ± SE | 3.37 ± 1.19 | 3.00 ± 0.86 | 2.807 ± 1.359 |
| Range | 1.88~5.24 | 1.87~4.95 | 1.01~5.53 |
| **Nitrite content [B]** | | | |
| Means ± SE | 1.12 ± 0.80 [b] | 4.80 ± 3.46 [a] | 2.57 ± 2.15 [b] |
| Range | 0.21~2.39 | 0.56~13.34 | 0.35~8.06 |
| **Peroxide value [C]** | | | |
| Means ± SE | 17.73 ± 17.22 | 13.59 ± 9.29 | 13.51 ± 6.70 |
| Range | 1.54~61.70 | 3.40~34.00 | 7.59~26.58 |

Results are expressed as means with standard errors including 14, 15, and 10 samples, respectively, and each sample data were means of three replicates. [A] Expressed as g/100 g muscle. [B] Expressed as mg/kg dry basis. [C] Expressed as meq $O_2$/kg of fat. [a,b] Means in the same row with different superscripts differ significantly ($p < 0.05$).

### 3.2. Characteristics of the Sequencing Data

As a result of quality control, an average of 77,311 high-quality sequences were obtained from each bacon sample (Table A1), which agreed with previous works in which the number of reads about each fermentation meat ranged from 33,983 to 170,428 [25]. All sequences were assembled to 1364 OTUs based on a 97% identity level. The indices of alpha diversity are shown in Table A2. Sufficient coverage of data was proven by up to 99% of the Good's coverage for all the samples. Although the amount of OTUs in each sample fluctuated obviously, ranging from 53 to 769, there was no difference between the basin, mountains, and plateau groups, owing to the total OTUs that were clustered to 1156, 1360, and 1298, respectively. In the meantime, the amount of shared OTUs among the three regions was up to 1101 (Figure 1), close to 81% of the total number of OTUs, indicating a high level of similarity in the bacteria compositions among the three groups. Traditional Sichuan bacon has developed its own style through long-term development, and the key formulations and manufacturing processes have remained almost unchanged since the Ming Dynasty, although a wide variety of bacons come from the food preferences of the different regions in Sichuan. This may have been the reason why the bacterial diversity of bacon in different regions was similar.

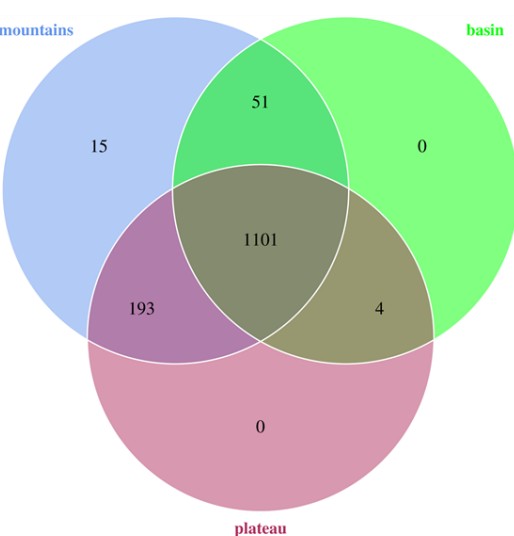

**Figure 1.** Venn diagrams of the three regions based on OTUs (3% distance). Basin: 14 bacons collected from Suining, Leshan, and Chengdu; mountains: 15 bacons collected from Yibin, Ya'an, and Xichang; plateau: 10 bacons collected from Kangding and Ma'er kang.

### 3.3. Bacterial Compositions of Sichuan Bacons

The relative abundances (%) of bacterial community proportions at the phylum and genus level are shown Figure 2. Firmicutes and Proteobacteria were the dominant phyla in all samples, ranging from 77.1% to 98.9%, and most samples revealed levels above 90%. Apart from these two phyla, Actinobacteria, Oxyphotobacteria, and Bacteroidetes were observed in some samples at levels above 1% (Figure 2A). This agrees with research on other Chinese traditional meat products such as Chinese sausage and ham [18,26]. At the genus level (Figure 2B), *Staphylococcus* was the dominant genus in the basin, mountains, and plateau group bacons, covering 26.7%, 20.6%, and 22.7% of the total genus, respectively. Conversely, the relative abundances of *Lactobacillus*, accounting for 0.5%, 3.5%, and 2.5%, respectively, were much lower than those of *Staphylococcus*. Many researchers support the belief that LAB and CNS are the most active microorganisms derived from meat fermentation [5]. The previous studies focused more on *Lactobacillus* diversity and function in Chinese traditional meat products [27], while little attention was paid to *Staphylococcus*. Moreover, the diversity of CNS in meat products is complex and variable because it is subject to multiple factors, including the type of raw muscle used, the ingredients added, and the processing conditions applied [28]. For instance, the pH increases due to the use of lactate during post-ripening, leading to replacement of the dominant species from *S. saprophyticus* to *S. equorum*, which is more adapted to weaker acidic environments [29]. It is reasonably believed that the role of *Staphylococcus* in the formation of the unique flavor and quality of Sichuan bacon should be paid more attention in future research.

In addition to *Staphylococcus*, some other predominant genera of relative abundance were observed in the samples. For instance, *Phyllobacterium* (16.8%), *Psychrobacter* (11.5%), *Brochothrix* (6.5%), *Vibrio* (6.3%), and *Delftia* (3.2%) were the abundant genera in the basin group, which are commonly spoilage bacteria in meat products [30]. In the plateau group, the relative abundance of *Pseudomonas* was up to 10.2%, which was much higher than the other two groups. This gram-negative bacteria, which relied on proteolytic activity, was a predominant spoilage bacteria in stored meat under low-temperature and aerobic conditions [31]. In the meantime, *Brucella* (5.1%), unidentified *Clostridiales* (3.7%), *Vibrio* (3.4%), *Lactococcus* (3.4%), and *Enterococcus* (3.2%) were observed in the plateau group, and *Phyllobacterium* (8.0%), *Lactobacillus* (3.5%), *Pseudomonas* (3.4%), and *Brucella* (3.1%) were observed in the mountains group. Undesirably, unidentified *Rickettsiales* (21%), which has never been found in bacon, was one of the dominant genera in the mountains group. The genus was derived from raw meat contamination, owing to this prokaryote only

parasitizing eukaryotic cells [32]. Although spontaneous fermented meat products are recognized as having excellent flavor, this can sometimes give rise to poor quality products, including decreased appearance quality, a shortened shelf life, and increased health risks. For this reason, the addition of starter cultures in traditional meat products is the best way to reduce potential risk caused by microorganisms and is also the best choice for Sichuan bacon.

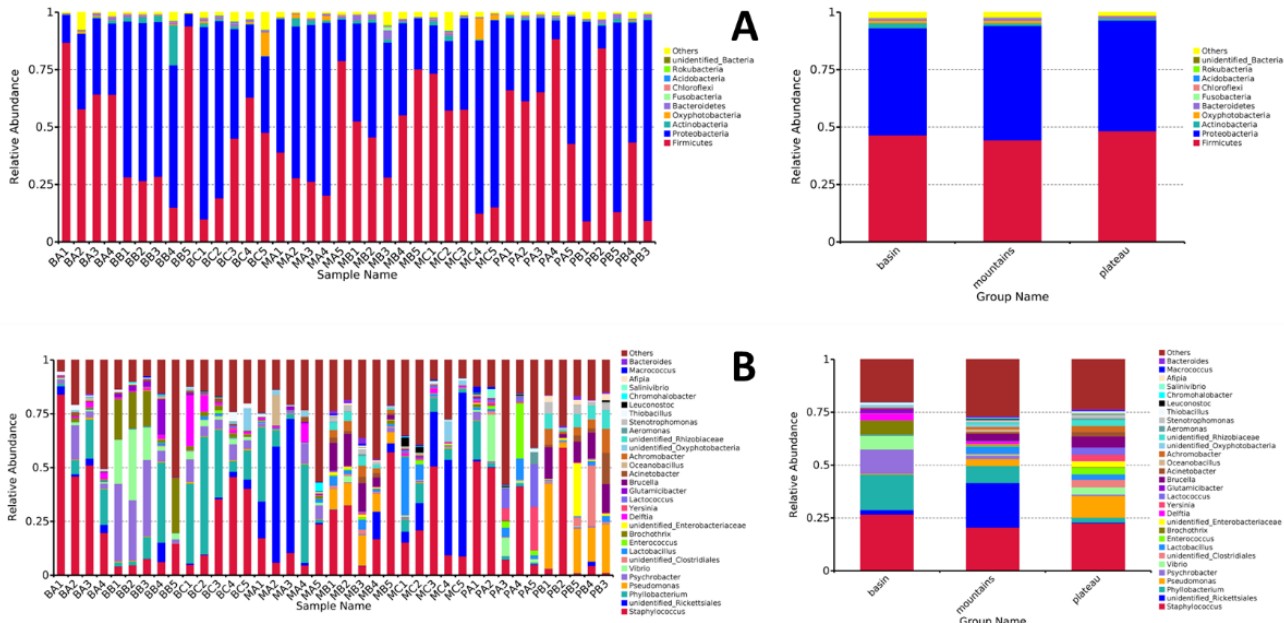

**Figure 2.** The relative abundances of bacterial compositions of different bacons (**A**) at the phylum level and (**B**) at the genus level.

### 3.4. Analysis of Bacterial Communities of Sichuan Bacon in Different Geographical Regions

The differences between the bacterial communities was compared in the samples collected from three regions of Sichuan to investigate the association between sampling location and bacterial communities (Figure 3). For PCA analysis (Figure 3A), there were 22.36% and 13.23% total variations in the first and the second components, respectively. PCA analysis showed that the samples collected from the basin region could be clustered together in small ranges. Interestingly, only a part of the mountains and plateau samples were clustered with the basin group, and the remaining samples were scattered long distance. PCoA (PC1 58.43% and PC2 15.32%, Figure 3B) and NMDS (stress: 0.081, Figure 3C) analysis based on the Weighted Unifrac distance showed a similar clustering pattern in the bacterial structures for the three groups. Moreover, a significant test of the community structure difference between groups based on the Anosim nonparametric test method of OTU relative abundance showed that there was a significant difference ($p < 0.05$, data not shown) between the basin and mountains, basin and plateau, mountains and plateau, or basin, mountains, and plateau. Considered together, the bacterial diversity of Sichuan bacon in different geographical regions showed a certain specificity as well as similarity. It is believed that long-term regular processes in spontaneous fermentation food often leads to a stable microbiota, among which only the best-adapted strains will occur [33]. On the contrary, microbial communities are strongly influenced by external environment conditions, such as a low environment temperature due to a production date from December of the current year to January of the following year, a high level of salt stresses due to formulation, or low water activity due to the long production cycle in meat ripening.

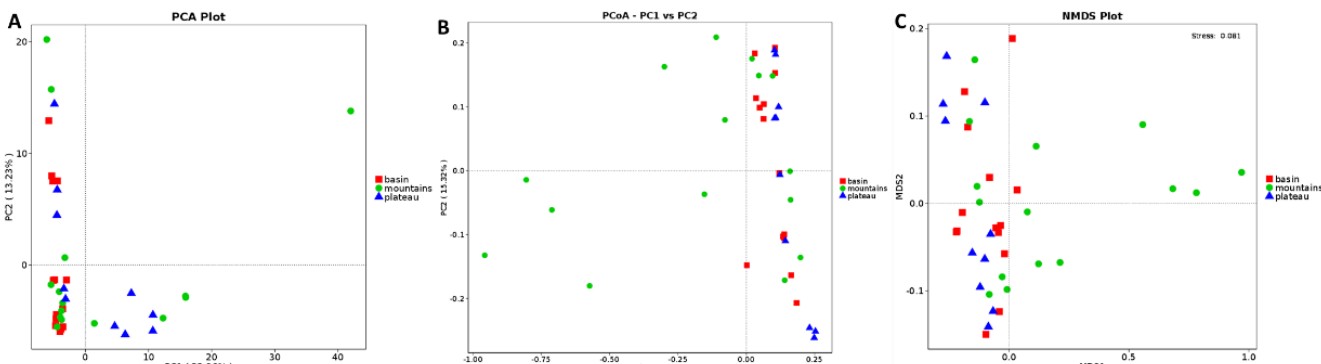

**Figure 3.** The beta diversity of the 39 bacons at the genus level (PCA (**A**), PCoA (**B**), and NMDS (**C**) scores plotted based on Weighted Unifrac).

LefSe analysis was able to identify statistically significant biomarkers, which were bacterial taxa about significant differences between the groups. The taxon nodes were displayed in terms of LDA scores >4 (Figure A1), and a few predominant genera are illustrated in Figure 4. The relative abundances of *Lactobacillus* in the mountains and plateau groups were much higher than those in the basin group (Figure 4A), while the opposite was true for *Staphylococcus*. The results showed that there was a negative correlation in the relative abundance between *Lactobacillus* and *Staphylococcus*, which is perhaps one of the most important factors in shaping the typical style of Sichuan bacon. Interestingly, the relative abundances of *Lactococcus* in the plateau group (Figure 4B), similar to *Enterococcus* (not listed), were much higher than those in the basin and mountains groups. These genera also can produce a large amount of organic acid, such as acetic and lactic acid, during exponential growth, following conversion to the alcohols and ketone that contribute a richer flavor and aroma to the product [34,35]. It is suggested that *Lactococcus* and *Enterococcus* are the predominant acid-producing bacteria rather than *Lactobacillus* in the plateau group. Moreover, the relative abundances of *Brochothrix* in the basin group were much higher than those in the mountains and plateau groups (Figure 4C), while the relative abundances of *Pseudomonas* from the basin group were much lower than those from the mountains and plateau groups (Figure 4D). Although there were undesirable microorganisms present in all samples of Sichuan bacon due to outdoor ripening processing, the undesirable microflora was different between regions, which means that different strategies should be prepared to ensure the safety of the products.

### 3.5. The Relationship between Top Genera and Environmental Factors

The commonly accepted view is that variables such as the Aw, pH, salt content, POV, and nitrite can induce changes in microbial communities in microbial ecosystems [36]. Five environmental factors were screened using VIF and BioENV tools, and the results demonstrated that the environmental factors had a significant impact on the microbiota. Parametric correspondence analysis (CCA, Figure 5A) and Spearman analysis (Figure 5B) were used to further investigate the relationship between the top genera (top 15 genera of relative abundance) and environmental factors. Obviously, the salt content and peroxide value had no significant ($p > 0.05$) effect on the bacterial community, owing to no significant difference between the three groups in this test. Aw was found to be positively correlated with the relative abundances of *Lactobacillus* and negatively correlated with the relative abundances of *Brochothrix*, *Vibrio*, *Delftia*, and *Psychrobacter*. The pH was negatively correlated with the relative abundances of *Delftia*, *Phyllobacterium*, and *Psychrobacter*. Nitrite was positively correlated with the relative abundances of unidentified Clostridiales, *Brucella*, *Lactococcus*, *Pseudomonas*, and unidentified enterobacteriaceae and negatively correlated with the relative abundances of *Psychrobacter*, *Brochothrix*, *Vibrio*, *Phyllobacterium*, and *Delftia*. The results showed that nitrite was the most important factor due to significant correlation

with many critical genera, followed by the Aw and pH. Nitrite plays a decisive role in cured meat products, providing a high-quality red color and flavor for products, especially for being recognized as bacteriostatic and bactericidal agents against pathogenic bacteria [37]. Nitrite that considered harmful to human health is not added in traditional Sichuan bacon, however its integrated functions are still difficult to replace in the meat industrialization. A combined strategy may be a better solution than a single agent. Interestingly, all three environmental factors were mostly negatively correlated with the undesired microorganisms in Sichuan bacon, which revealed that an appropriate level of Aw, pH, and nitrite plays a vital role in the safety of Sichuan bacons.

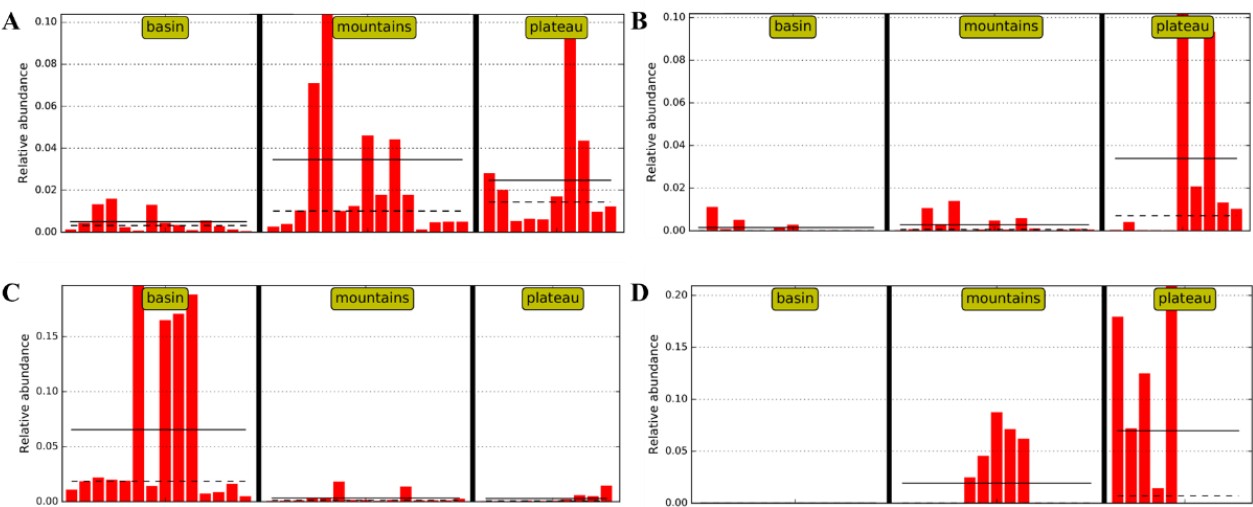

**Figure 4.** Significantly different genera through LefSe analysis of different abundant taxons among three regions of Sichuan bacon: (**A**) *Lactobacillus*, (**B**) *Lactococcus*, (**C**) *Brochothrix*, and (**D**) *Pseudomonas*.

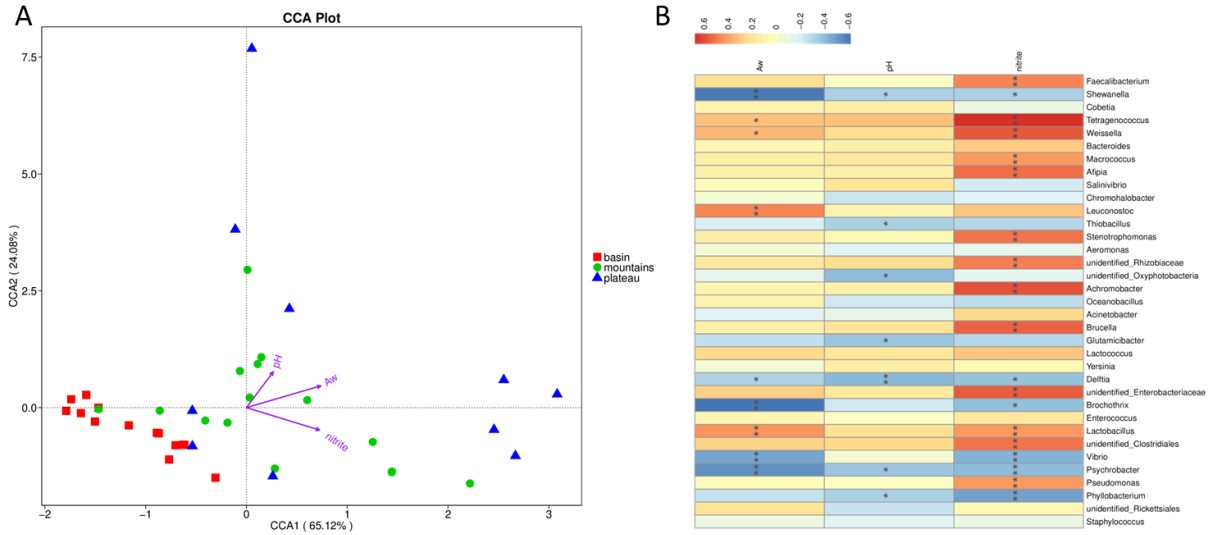

**Figure 5.** The parametric correspondence analysis (**A**) and Spearman analysis (**B**) of the top genera and environmental factors among the three regions.

## 4. Conclusions

In conclusion, there was a large number of undesired microorganisms in Sichuan bacon collected from three geographical regions, which was significantly correlated with the physiochemical characteristics. Our data suggested that *Staphylococcus* played the most important role in Sichuan bacon, and *Lactococcus* and *Enterococcus* were more important in the plateau group than in the other two groups. There were significant differences in

undesired bacterial community structures among the geographical groups, which were also affected by physiochemical parameters such as the Aw, pH, and nitrite levels. This may indicate that a variety of control strategies should be applied to Sichuan bacon in different geographical regions. Furthermore, more studies regarding the species and strain levels of bacterial diversity in Sichuan bacon by metagenomics and culturomics are needed. Moreover, fermentation starters consisting of functional strains to improve the safety and quality of industrial products are also necessary.

**Author Contributions:** Formal analysis, visualization, and writing–original draft preparation, S.W.; formal analysis, investigation, and resources, X.W.; investigation and resources, W.P.; writing—reviewing and editing, A.L.; conceptualization, methodology, supervision, and funding acquisition, S.L.; conceptualization, methodology, and supervision, Y.Y.; methodology and supervision, L.Z. All authors have read and agreed to the published version of the manuscript.

**Funding:** This research was funded by financial support from the Science & Technology Foundation of Chengdu, grant number 2019YF0900050SN, and by the National Natural Science Foundation of China, grant number 31701518.

**Institutional Review Board Statement:** Not applicable.

**Informed Consent Statement:** Not applicable.

**Data Availability Statement:** The data presented in this study are available on request from the corresponding author.

**Conflicts of Interest:** The authors declare no conflict of interest.

### Appendix A

**Table A1.** The number of sequences for each sample.

| Sample | Raw PE (#) | Combined (#) | Qualified (#) | Nochime (#) | Avg Len (nt) | Effective % |
|--------|-----------|--------------|---------------|-------------|--------------|-------------|
| BA1 | 98,835 | 96,773 | 95,663 | 93,267 | 252 | 94.37 |
| BA2 | 87,792 | 63,104 | 60,343 | 57,774 | 254 | 65.81 |
| BA3 | 86,549 | 84,950 | 83,974 | 80,927 | 252 | 93.5 |
| BA4 | 83,900 | 79,274 | 77,945 | 75,347 | 252 | 89.81 |
| BB1 | 84,898 | 82,385 | 81,136 | 73,781 | 253 | 86.91 |
| BB2 | 86,495 | 83,717 | 82,557 | 76,913 | 253 | 88.92 |
| BB3 | 95,010 | 90,558 | 89,135 | 84,040 | 253 | 88.45 |
| BB4 | 92,873 | 83,191 | 81,768 | 76,352 | 249 | 82.21 |
| BB5 | 87,069 | 75,142 | 73,213 | 70,254 | 254 | 80.69 |
| BC1 | 81,467 | 73,264 | 70,411 | 66,509 | 252 | 81.64 |
| BC2 | 98,199 | 94,404 | 93,045 | 87,076 | 253 | 88.67 |
| BC3 | 85,961 | 76,897 | 75,270 | 71,158 | 253 | 82.78 |
| BC4 | 94,842 | 80,522 | 77,919 | 72,685 | 253 | 76.64 |
| BC5 | 95,153 | 76,999 | 74,828 | 71,166 | 252 | 74.79 |
| MA1 | 82,378 | 76,064 | 74,786 | 72,287 | 249 | 87.75 |
| MA2 | 85,426 | 83,136 | 82,413 | 79,576 | 237 | 93.15 |
| MA3 | 97,461 | 88,505 | 87,206 | 85,441 | 234 | 87.67 |
| MA4 | 83,188 | 82,167 | 81,227 | 76,744 | 253 | 92.25 |
| MA5 | 92,712 | 87,305 | 85,919 | 82,609 | 253 | 89.1 |
| MB1 | 82,432 | 79,911 | 79,157 | 73,762 | 254 | 89.48 |
| MB2 | 89,052 | 84,857 | 83,635 | 79,450 | 254 | 89.22 |
| MB3 | 76,775 | 73,559 | 71,681 | 66,512 | 255 | 86.63 |
| MB4 | 88,555 | 85,598 | 83,973 | 80,118 | 250 | 90.47 |
| MB5 | 93,716 | 91,280 | 90,427 | 87,884 | 252 | 93.78 |
| MC1 | 92,442 | 87,084 | 85,654 | 79,627 | 252 | 86.14 |
| MC2 | 98,422 | 82,567 | 80,554 | 76,418 | 250 | 77.64 |
| MC3 | 84,825 | 80,792 | 79,890 | 77,608 | 245 | 91.49 |
| MC4 | 99,950 | 95,372 | 94,375 | 91,363 | 239 | 91.41 |

**Table A1.** *Cont.*

| Sample | Raw PE (#) | Combined (#) | Qualified (#) | Nochime (#) | Avg Len (nt) | Effective % |
|---|---|---|---|---|---|---|
| MC5 | 90,170 | 88,416 | 87,902 | 86,234 | 229 | 95.63 |
| PA1 | 89,505 | 85,058 | 83,807 | 80,253 | 253 | 89.66 |
| PA2 | 96,874 | 88,771 | 86,815 | 81,797 | 253 | 84.44 |
| PA3 | 98,694 | 95,418 | 93,892 | 86,847 | 253 | 88 |
| PA4 | 86,633 | 82,189 | 80,884 | 76,942 | 253 | 88.81 |
| PA5 | 87,876 | 70,756 | 68,491 | 65,280 | 254 | 74.29 |
| PB1 | 80,545 | 77,321 | 76,581 | 74,041 | 253 | 91.93 |
| PB2 | 77,140 | 74,767 | 72,543 | 69,946 | 254 | 90.67 |
| PB3 | 88,085 | 84,479 | 82,851 | 77,546 | 253 | 88.04 |
| PB4 | 88,531 | 86,156 | 84,593 | 80,461 | 253 | 90.88 |
| PB5 | 76,757 | 74,594 | 73,210 | 69,141 | 254 | 90.08 |

All data come from the report of NOVOGENE. The (#) represents the number of sequences and the (nt) represents average base-pairs number of effective data.

**Table A2.** Alpha diversity indices of each sample.

| Sample | Observed Species | Shannon | Simpson | Chao1 | Ace | Good's Coverage |
|---|---|---|---|---|---|---|
| BA1 | 346 | 1.5303 | 0.3083 | 357.413 | 368.0392 | 0.9991 |
| BA2 | 281 | 3.2806 | 0.7633 | 277.4737 | 279.2758 | 0.9998 |
| BA3 | 357 | 2.9900 | 0.7031 | 358.4685 | 385.1882 | 0.9993 |
| BA4 | 448 | 3.1576 | 0.7523 | 463.6195 | 475.4967 | 0.9994 |
| BB1 | 306 | 3.6041 | 0.8341 | 312.2462 | 334.4232 | 0.9996 |
| BB2 | 349 | 3.4318 | 0.8120 | 362.2581 | 367.6659 | 0.9994 |
| BB3 | 389 | 3.6992 | 0.8556 | 400.7222 | 419.8041 | 0.9993 |
| BB4 | 749 | 4.1670 | 0.8242 | 755.6667 | 769.4436 | 0.9988 |
| BB5 | 53 | 1.8340 | 0.6303 | 52.88235 | 58.51154 | 0.9999 |
| BC1 | 680 | 3.8438 | 0.7982 | 696.4286 | 708.2509 | 0.9991 |
| BC2 | 683 | 3.3852 | 0.6842 | 683.7653 | 710.7533 | 0.9991 |
| BC3 | 706 | 3.7436 | 0.7757 | 713.7213 | 725.2032 | 0.9993 |
| BC4 | 557 | 3.8612 | 0.7721 | 565.3404 | 577.2322 | 0.9993 |
| BC5 | 534 | 3.6825 | 0.8016 | 562.6241 | 567.1605 | 0.9989 |
| MA1 | 284 | 3.0673 | 0.7926 | 294.9014 | 294.3744 | 0.9995 |
| MA2 | 312 | 3.6688 | 0.8350 | 321.0851 | 327.6277 | 0.9993 |
| MA3 | 377 | 2.8811 | 0.6826 | 379.6726 | 375.6161 | 0.9994 |
| MA4 | 511 | 3.1646 | 0.7491 | 513.4286 | 540.6371 | 0.9992 |
| MA5 | 374 | 3.1296 | 0.7168 | 372.0435 | 387.8292 | 0.9994 |
| MB1 | 610 | 4.8808 | 0.8810 | 608.28 | 633.0211 | 0.9993 |
| MB2 | 594 | 4.5379 | 0.8608 | 609.1866 | 612.6939 | 0.9995 |
| MB3 | 740 | 6.3633 | 0.9614 | 740.6842 | 740.852 | 0.9996 |
| MB4 | 617 | 4.5869 | 0.8680 | 627.4681 | 640.6145 | 0.9994 |
| MB5 | 476 | 3.1935 | 0.6678 | 501.5 | 497.8253 | 0.9989 |
| MC1 | 706 | 4.3571 | 0.8640 | 705.52 | 725.3433 | 0.9992 |
| MC2 | 769 | 4.5736 | 0.8899 | 772.4904 | 792.9436 | 0.9991 |
| MC3 | 464 | 3.1463 | 0.7101 | 475.0074 | 485.7014 | 0.9992 |
| MC4 | 470 | 3.5864 | 0.8186 | 498.1161 | 513.0337 | 0.9986 |
| MC5 | 368 | 2.2450 | 0.5116 | 386.7835 | 419.8829 | 0.9991 |
| PA1 | 363 | 3.0520 | 0.6912 | 362.2222 | 375.9001 | 0.9995 |
| PA2 | 646 | 3.2985 | 0.7065 | 642.3623 | 664.7128 | 0.9991 |
| PA3 | 544 | 3.8095 | 0.8344 | 571.2261 | 579.1615 | 0.9990 |
| PA4 | 646 | 3.3790 | 0.7589 | 637.9827 | 669.531 | 0.9992 |
| PA5 | 382 | 3.7926 | 0.8719 | 398.703 | 414.029 | 0.9992 |
| PB1 | 544 | 4.1810 | 0.8650 | 544.175 | 563.6767 | 0.9993 |
| PB2 | 416 | 2.6434 | 0.6151 | 436.2581 | 435.0155 | 0.9993 |
| PB3 | 539 | 4.5015 | 0.9082 | 554.5118 | 559.1409 | 0.9992 |

**Table A2.** *Cont.*

| Sample | Observed Species | Shannon | Simpson | Chao1 | Ace | Good's Coverage |
|---|---|---|---|---|---|---|
| PB4 | 522 | 4.4449 | 0.8858 | 533.8261 | 533.6284 | 0.9994 |
| PB5 | 474 | 4.6674 | 0.9015 | 491.8583 | 493.4462 | 0.9993 |

All data come from the report of NOVOGENE.

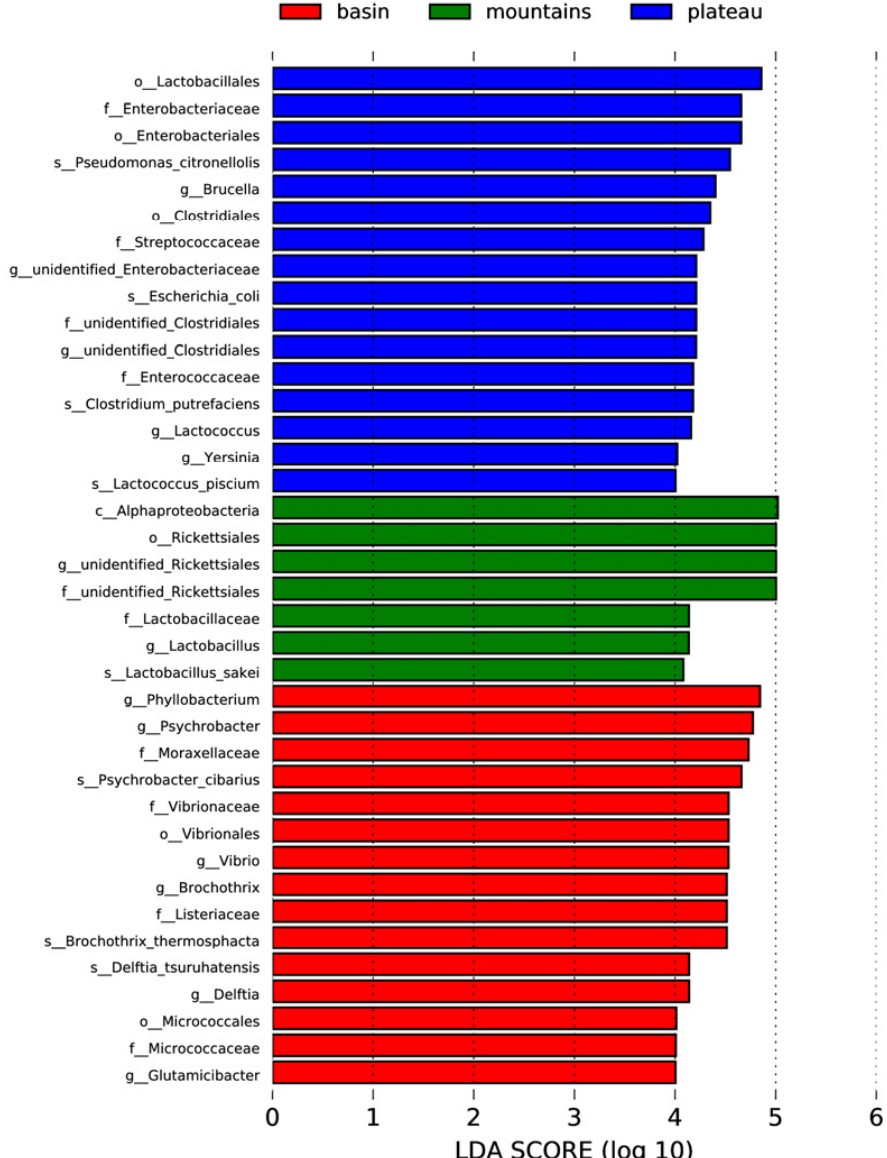

**Figure A1.** Significantly discriminant taxa (LDA score >4) among three regions of Sichuan bacon by LefSe analysis.

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
