# Peer review of "Evaluation of Bacterial Diversity and Quality Features of Traditional Sichuan Bacon from Different Geographical Region"

_applsci, doi:10.3390/app11209738_

Round 1

Reviewer 1 Report

The work carried out is a valuable contribution to understand the role of bacterial communities in food products particularly with regard to the limited number of published contributions available in this regard.

However, the section materials and methods is not adequately described and the section conclusions is not concrete enough. Another shortcoming of the study is that no alternative methods were used to verify the results (microbiological tests, etc.).

The following changes should be made:

  1. Materials and methods

More precision in the information is needed in the whole section.

- Are the samples from one year of production? If not, how were the seasonal influences taken into account? How was the originality of the samples confirmed. Please specify.

- In some cases, essential details on the methodology used are missing or the citation is not available (111: Aw value; 117ff: amplicon sequencing, which platform, marker-system/primer information, PCR-conditions or the citation),

- in some cases the information is too vague (122ff: "Effective tags were obtained...", How was an effective tag identified? Please specify and add details on how data quality was checked. details on DNA-extraction method is missing; 123: "The Mothur method...", "The Mothur method" does not exist, please specifiy?

- Details on the replicates are not always comprehensible (136: "All analyses were run in triplicate.". What do you mean with all (from sampling onwards)? Please specify.

  1. Results

183: As a result of quality control, an average of 77,311 high-quality sequences….”:

How was the quality control carried out? How are high-quality sequences defined? Please specify.

197: “Maybe this was the reason why the bacterial diversity tended to be consistent with sufficient samples (NO. ≥ 10) in different regions”:

This is not clear enough, please rephrase. What NO. > 10 mean in this context?

275ff: Results of LefSe analysis (Fig.A.1.) is not clear enough, please add some details on biomarkers identified.

Table A1 – sample BB5: what is your explanation about the extremely low number of observed species compared to the rest of the data provided?

Table A2 – sample BA2/BC4/BC5: This samples show a prominent loss of reads during data analysis. Is it only due to bad quality of raw data? Have you checked the raw data (including the sorted out reads) to see if there are any discrepancies?

  1. conclusions

The conclusions are not concrete enough.

331ff: “In conclusion, the bacterial community and physiochemical characteristics of Sichuan bacon collected from three different geographical regions have been well analyzed.”: Rephrasing necessary, be more concrete (hygiene aspects, diversity aspects). Further activities should include alternative approaches and more data on species level, even if information on starter cultures are necessary.

----------------

79: there ist not only one HTS-method, please rephrase.

240ff: Undesirably, unidentified Rickettsiales (21%), never been found in bacon, was one of the dominant genus in the mountains-group. The genus derived from raw meat of contamination owing to this prokaryote can only parasitize on eukaryotic cells (Aurélie et al., 2011. Please change as indicated.

Reviewer 2 Report

The authors report on the bacterial diversity of a type of bacon originating from 3 different regions in China. Their methodology, analytical procedures and statistical representation is of high quality, and results provide further insight into microbial community interaction and food safety. This is also a good example of the scientific  basis of cultural and geographical traditions. 

In the methods section a brief description of the extraction (Yang et al 2018) procedures, and further reference or description of the analytical (NOVOGENE) techniques should be added.

The manuscript is well written. However, to correct frequent linguistic errors, I recommend proof-reading by an English speaking scientist. 
